# Comparison of Gonadal Transcriptomes Uncovers Reproduction-Related Genes with Sexually Dimorphic Expression Patterns in *Diodon hystrix*

**DOI:** 10.3390/ani11041042

**Published:** 2021-04-07

**Authors:** Huapu Chen, Zhiyuan Li, Yaorong Wang, Hai Huang, Xuewei Yang, Shuangfei Li, Wei Yang, Guangli Li

**Affiliations:** 1Guangdong Research Center on Reproductive Control and Breeding Technology of Indigenous Valuable Fish Species, Key Laboratory of Marine Ecology and Aquaculture Environment of Zhanjiang, Fisheries College, Guangdong Ocean University, Zhanjiang 524088, China; chenhp@gdou.edu.cn (H.C.); g_yl903@163.com (Z.L.); yaorongwang217@126.com (Y.W.); 2Key Laboratory of Utilization and Conservation for Tropical Marine Bioresources, Hainan Tropical Ocean University, Ministry of Education, Sanya 572022, China; huanghai74@126.com; 3Shenzhen Key Laboratory of Marine Bioresource and Eco-Environmental Science, College of Life Sciences and Oceanography, Shenzhen University, Shenzhen 518060, China; yangxw@szu.edu.cn (X.Y.); szu_sfli@163.com (S.L.); 4Food and Environmental Engineering Department, Yangjiang Polytechnic, Yangjiang 529566, China

**Keywords:** *Diodon hystrix*, transcriptome, gonad, reproduction, sex-biased genes

## Abstract

**Simple Summary:**

Spot-fin porcupine fish (*Diodon hystrix*) has been recognized as a new and emerging aquaculture species with promising economic value in south China. However, due to the lack of understanding of reproductive regulation, the management of breeding and reproduction under captivity remains a major technological barrier for the development of large-scale aquaculture of *D. hystrix*. In this study, the first gonad transcriptomes of *D. hystrix* were analyzed using Illumina HiSeq sequencing. Comparison of ovary and testis transcriptomes identified a set of differentially expressed genes (DEGs) supposed to be associated with gonadal development and gametogenesis. The conserved expression profiles of the well-known reproduction-related genes implies their similar roles in gonad differentiation and development in *D. hystrix*. The detailed transcriptome data can improve our understanding of the regulatory functions of sex-related genes in *D. hystrix*.

**Abstract:**

*Diodon hystrix* is a new and emerging aquaculture species in south China. However, due to the lack of understanding of reproductive regulation, the management of breeding and reproduction under captivity remains a barrier for the commercial aquaculture of *D. hystrix*. More genetic information is needed to identify genes critical for gonadal development. Here, the first gonadal transcriptomes of *D. hystrix* were analyzed and 151.89 million clean reads were generated. All reads were assembled into 57,077 unigenes, and 24,574 could be annotated. By comparing the gonad transcriptomes, 11,487 differentially expressed genes were obtained, of which 4599 were upregulated and 6888 were downregulated in the ovaries. Using enrichment analyses, many functional pathways were found to be associated with reproduction regulation. A set of sex-biased genes putatively involved in gonad development and gametogenesis were identified and their sexually dimorphic expression patterns were characterized. The detailed transcriptomic data provide a useful resource for further research on *D. hystrix* reproductive manipulation.

## 1. Introduction

Spot-fin porcupine fish (*Diodon hystrix*) belonging to the family Diodontidae (Teleostei, Tetraodontiformes) is mainly distributed in the lagoons and coral reefs of tropical seawater [1,2]. Although *D. hystrix* is widely considered to have tetrodotoxin [3], this fish species is noted for its delicious meat and nutritious skin that is rich in collagen, especially in the Pacific Islands region and southern Hainan Island, China [4,5]. According to the assessment of nutrient value, *D. hystrix* has a high content of amino acids and the rate of essential amino acids is accorded with the FAO/WHO standard. Moreover, the fat contents in skin and muscle tissues are found to be negligible, contributing to a low energy value [6]. *D. hystrix* with such distinguished features is increasingly becoming a highly popular food fish with considerable economic value, the commercial demand is rising rapidly in south China and the market price continues to climb [5,6].

The outstanding growth performance and ease of cultivation make *D. hystrix* a very promising species for commercial aquaculture [6]. As a gonochoristic fish species, however, porcupine fish gonad maturates earlier in males than females under either natural or aquaculture conditions. It is, therefore, difficult, to obtain fertilized eggs using conventional techniques of artificial propagation. With the bloom of porcupine fish industry, artificial breeding unfortunately remains a significant barrier for the successful aquaculture of *D. hystrix*. The commodity fish still has to be captured from the sea. Due to a weak resilience of the natural population of *D. hystrix*, its wild stock is now gravely threatened and the fish resource has been declining greatly in recent years as a result of critical hazards, such as a sharp increase of fish catch and continued destruction of natural habitat [7]. Thus, the large-scale aquaculture of *D. hystrix* is absolutely in great and urgent need of artificial breeding and reproduction management.

The reproduction process, typically including gonad differentiation, development, maturation and gametogenesis, is more complex in fish than in other vertebrate species and is under the influence of genetic and environmental factors [8,9]. Years of practical experience have shown that a thorough knowledge of the mechanisms involved in reproductive regulation could strongly promote the development of efficient management of reproduction, which is a necessary prerequisite for the breeding of aquaculture species. To date, however, the only studies in *D. hystrix* focus mainly on systematics and zoogeography [2,4], characterization of mitochondrial genome and phylogenetic analysis [5], molecular marker development [7], parasitic diseases [10,11], tetrodotoxic poisoning mechanism [3], as well as the analysis of nutrient compositions that can provide reference data for the formulation design of special diets [6]. The research efforts on reproductive biology are severely limited and the underlying mechanism of reproduction remains poorly understood. These research gaps seriously impede the progress of artificial breeding techniques for *D. hystrix*. To provide useful guidance to the future practice of reproductive manipulation, it is essential to pay serious attention to delineating the regulatory mechanisms of the reproductive process. In particular, the efforts to functionally explore the key genes associated with sex differentiation, gonad development and maturation, and gametogenesis are indispensable.

Compared with other commercial aquaculture species, the genetic information available for *D. hystrix* is rather limited. Thus far, reproduction-related genes have been rarely reported in *D. hystrix* and a fundamental understanding of their expression profiles is still lacking. Before elucidating the specific roles of these genes, there is an immediate need to enrich the genomic background knowledge. With improved efficiency, next-generation sequencing (NGS) based transcriptome sequencing is an efficient gene expression profiling technology that is superior in generating a great amount of transcript sequences and mRNA expression data rapidly and cost-effectively, especially for non-model species [12,13]. It has been frequently employed in functional gene identification and gene-expression regulation analysis in aquaculture fish species [14,15,16] to provide a general representation of the genes that are expressed in specific tissues [17]. The expression patterns of reproduction-related genes generally exhibit significant differences between the sexes both in the developing and developed gonads. By comparative analysis of gonad transcriptomes, many candidate genes and pathways involved in sexual or reproductive regulation and gonad maturation have been identified in silver sillago (*Sillago sihama*) [18], spotted knifejaw (*Oplegnathus punctatus*) [19], olive flounder (*Paralichthys olivaceus*) [20], yellow catfish (*Pelteobagrus fulvidraco*) [21], Amur catfish (*Silurus asotus*) [22], and Amur sturgeon (*Acipenser schrenckii*) [23]. These studies provide helpful insights into reproduction-related genes and enable the discovery of new gene candidates.

In this study, Illumina-based gonadal RNA sequencing, Trinity de novo assembly and annotation were firstly conducted in *D. hystrix*. Furthermore, comparative transcriptomics was applied to reveal the expression patterns of sex-biased genes and the differences in the expression of the genes which potentially are involved in the regulation of reproduction were analyzed and discussed. This study mainly aimed i) to further enrich the available genetic and genomic data for a deeper understanding of gene expression and functional gene mining, and ii) to identify as many genes putatively related to gonad differentiation, development, maturation, and gametogenesis as possible for future research into the molecular mechanisms of reproduction in *D. hystrix*.

## 2. Materials and Methods

### 2.1. Ethics Statement

Animal experiments were carried out in strict accordance with the recommendations in the Guide for the Care and Use of Laboratory Animals. The protocol was approved by the Animal Research and Ethics Committee of Guangdong Ocean University (NIH Pub. No. 85–23, revised 1996). All efforts were made to minimize animal suffering and to reduce the number of animals used.

### 2.2. Sample Collection and Preparation

For transcriptome sequencing, six adult *D. hystrix* (three males and three females) were obtained from the South China Sea (18°11′2.73′′ N, 109°18′32.66′′ E) on 12 May 2020 (see Appendix A for the specific information of the fish samples). Live fish were sacrificed by decapitation following anesthetization with a 300 mg/L tricaine methanesulfonate (MS222, Sigma, Saint Louis, MO, USA) immersion bath. After dissection, the determination of fish gender was performed by morphological observation of gonads. The gonad tissues were excised from *D. hystrix* individuals within 1 min from sacrifice, immediately quick-frozen in liquid nitrogen, and then stored at −80 °C until RNA extraction.

### 2.3. Illumina RNA Sequencing

Six gonad samples (three replicates each sex) were used for the preparation of transcriptome sequencing libraries. The RNA-Seq process was performed as described previously [24]. In brief, total RNA was isolated from female and male *D. hystrix* gonad tissues using a Trizol reagent kit (Life Technologies, Carlsbad, CA, USA). The isolated RNA was quantified by a Nanodrop 2000c spectrophotometer (Thermo Scientific, Wilmington, DE, USA), and its integrity was confirmed by agarose gel electrophoresis and Agilent 2100 BioAnalyzer System (Agilent Technologies, Santa Clara, CA, USA). After purifying mRNA with an Oligo-dT Beads Kit (Qiagen, Hilden, Germany), cDNA libraries were constructed using a TruSeq^®^ Stranded mRNA Sample Preparation kit following the manufacturer’s protocol. RNA sequencing of the libraries was performed using the Illumina HiSeq™ 2000 platform (Illumina, Inc., San Diego, CA, USA) that generates paired-end (PE) reads of 125 bp length.

### 2.4. De Novo Assembly

By means of SOAPnuke (version 1.5.0) [25], the raw reads were pruned using the software’s quality control with the parameters “-l 10 -q 0.5 -n 0.05 -p 1 -i”. In this step, clean data were generated by removing adapter sequences, reads containing ploy-N sequences and low-quality reads from the raw data. Then, the clean data were de novo assembled by Trinity RNA-Seq Assembler (version r20140717, http://trinityrnaseq.sourceforge.net (accessed on 15 June 2015)) with default parameters [26]. The shorter redundant final linear transcripts were eliminated using CD-HIT-EST when the sequences were totally covered by other transcripts with 100% identity, and the longest ones were defined as unigenes [24].

### 2.5. Annotation and Classification

Annotation was conducted by aligning sequence data against public databases using BLAST 2.2.26+ software (https://blast.ncbi.nlm.nih.gov/Blast.cgi (accessed on 20 April 2016)) with an *E*-value threshold of 1e-5. The unigenes were subjected to the sequence homology searches against the National Center for Biotechnology Information (NCBI) non-redundant (Nr), Protein family (Pfam), Clusters of Orthologous Groups of proteins (KOG/COG/eggNOG), Swiss-Prot, Kyoto Encyclopedia of Genes and Genomes (KEGG) databases. Further analysis was performed to obtain the Gene Ontology (GO) functions using the Blast2GO package [27]. The classification of GO terms was visualized using WEGO statistical software [28]. Additionally, KOBAS v2.0 (http://kobas.cbi.pku.edu.cn/home.do (accessed on 24 July 2015)) was employed to analyze the KEGG pathway annotation data and to obtain the pathway categories [29].

### 2.6. Differential Expression Analysis and Functional Enrichment

By means of the expected number of fragments per kb per million reads (FPKM) method, gene expression levels were calculated using RSEM software (version 1.2.15) [30]. The DESeq2 package was used to identify differentially expressed genes (DEGs) between ovaries and testes [31]. FDR value ≤ 0.01 and |log_2_ (Fold Change)| ≥ 1 were used as the threshold for significantly differential expression. Additionally, GO and KEGG functional enrichment analyses were performed to determine the DEGs that were significantly enriched in GO terms and KEGG pathways at Bonferroni-corrected *p*-value ≤ 0.05 compared with the whole-transcriptome background. GO enrichment analysis of DEGs was implemented by the topGO package’s (version 2.28.0) Kolmogorov–Smirnov test [32]. Finally, KOBAS v2.0 was used to test the statistical enrichment of DEGs in KEGG pathways [33].

### 2.7. Validation of DEGs by Real-Time Quantitative PCR (RT-qPCR)

A total of 23 DEGs putatively associated with reproduction were chosen to validate the results of RNA-seq by RT-qPCR analysis. The primer pairs are listed in Appendix A. Total RNA was isolated from gonad samples using TRIzol according to the manufacturer’s instructions (Invitrogen, Carlsbad, CA, USA). The RNA was then subjected to reverse transcription using a RevertAid first-strand cDNA synthesis kit (Fermentas, Vilnius, Lithuania). RT-qPCR was performed on an ABI 7500 qPCR system (Life Technologies Inc., Carlsbad, CA, USA) using SYBR Green Real Time PCR Master Mix (TaKaRa Biotechnology, Dalian, China). The reference gene β-actin was used as an internal control to determine the relative expression. Three independent biological replicates and two technique repeats were performed for each gene. The relative gene expression levels were calculated using 2^−ΔΔ*C*t^ method. Analysis of Variance (ANOVA) was performed by SPSS 17.0 (SPSS Inc., Chicago, IL, USA). Values with *p* < 0.05 were considered significant.

## 3. Results

### 3.1. Overview of Sequencing and Assembly Results

RNA-Seq of the six libraries produced a total of 156.58 million raw reads (46.85 Gb sequencing data) with a mean of 26.10 million, ranging from 21.49 to 40.74 million per sample (Table 1). Approximately 151.89 (97.00%) million clean reads with a mean Q30 of 94.32% were filtered from the raw data (Table 1). The total size of the clean data generated from each library reached more than 6.0 Gb. The statistics of sequencing saturation distribution and gene coverage showed that the sequencing coverage was sufficient to quantitatively analyze the gene expression profiles (Appendix A). All raw sequencing data were submitted to the Sequence Read Archive (SRA, http://www.ncbi.nlm.nih.gov/sra/ (accessed on 25 October 2020)) of the NCBI database under BioProject accession number PRJNA674446.

All clean data were then imported to the Trinity package for de novo assembly using the default parameters. The high-quality clean reads were assembled into 139,628 transcripts with a N50 length of 3118 bp (Table 2). Further redundancy elimination resulted in a total of 57,077 unigenes with an average length of 1300 bp. In terms of sequence length distribution, 33,675 (59.00%) unigenes were > 500 bp in length and 19,620 (34.37%) unigenes were >1000 bp in length (Table 2). These results demonstrated the high quality of assembly.

### 3.2. Unigenes Annotation

Functional annotation was carried out by aligning the 57,077 unigenes to the protein sequences of public databases. A total of 24,574 (43.05%) unigenes were successfully annotated in at least one of the queried databases (Table 3). Of which, 23,114 (40.50%) and 13,488 (23.63%) unigenes had homologous sequences in the Nr and Swiss-prot protein databases, respectively. Meanwhile, 12,434 (21.78%), 15,157 (26.56%), and 13,699 (24.00%) unigenes could be annotated and classified in the GO, KOG, and KEGG databases, respectively (Table 3). Specifically, 12,434 unigenes annotated in the GO database were divided into three main subcategories and assigned into 56 2nd GO terms (Appendix A), and 15,157 unigenes were successfully clustered into 25 KOG subcategories (Appendix A). Furthermore, 13,699 (24.28%) unigenes were grouped into six main KEGG categories that include 287 secondary pathways (Appendix A).

### 3.3. Differential Expression Analysis

The levels of gene expression were normalized using the FPKM values. The distribution of gene expression levels was shown as a series of box plots (Figure 1A). About half of the genes were found to be expressed at extremely low levels (0.1 ≤ FPKMs ≤ 1) or not expressed at all (0 ≤ FPKMs < 0.1) in the gonads, and only a small proportion was considered to be highly expressed (FPKMs ≥ 60). To ensure if all gonad samples were reliable, the gene expression patterns of testes and ovaries were visualized via a heatmap using the Pearson correlation coefficient as distance measure (Figure 1B). The correlation analysis clearly indicated that the samples were classified into two major groups, representing testis and ovary, respectively. In addition, the result of principal component analysis (PCA) showed that three testis samples (blue circles) formed a cluster and three ovary samples (red circles) formed another distinct cluster (Appendix A).

By comparison of the unigene expression levels in gonadal transcriptomes, a total of 11,487 unigenes were detected to be differentially expressed between the ovaries and testes (FDR ≤ 0.01, |Log_2_ (fold change)| ≥ 1) (Appendix A). Among these DEGs, 6888 testis-biased (downregulated) and 4599 ovary-biased (upregulated) transcripts with significant differences in expression levels between the sexes were obtained (Figure 2A). The further analysis indicated that 46 and 1163 DEGs were specifically expressed (0 ≤ FPKMs ≤ 0.1) in the ovaries and testes, respectively. The remaining 10,278 DEGs were expressed in both gonads (Figure 2B).

### 3.4. Enrichment Analysis of DEGs

GO functional analysis was performed and the obtained DEGs were finally assigned to 52 2nd level GO terms. Of these functional terms, ‘cellular process’, ‘binding’, ‘single-organism process’, and ‘metabolic process’ annotated the most DEGs (Figure 3). More important, many GO terms associated with reproduction were identified, such as ‘reproductive process’ (GO:0022414), ‘gamete generation’ (GO:0007276), ‘steroid hormone mediated signaling pathway’ (GO:0043401), and ‘oocyte differentiation’ (GO:0009994) (Appendix A). GO enrichment analysis showed that 224, 62, and 107 GO terms were significantly enriched in categories ‘biological process’, ‘cellular component’, and ‘molecular function’, respectively (*p* < 0.05) (Appendix A). The top three most significant GO terms involved in ‘biological processes’ were ‘DNA repair’ (GO:0006281), ‘tRNA processing’ (GO:0008033), and ‘lymph vessel development’ (GO:0001945). The top three most significant terms involved in ‘molecular functions’ included ‘RNA methyltransferase activity’ (GO:0008173), ‘DNA binding’ (GO:0003677), and ‘chemorepellent activity’ (GO:0045499). The top three ‘cellular components’ GO terms were ‘catalytic complex’ (GO:1902494), ‘ribosome’ (GO:0005840), and ‘cytoplasm’ (GO:0005737).

KEGG enrichment analysis was also carried out with the DEGs to uncover their functional characteristics. In total, 5598 DEGs were mapped to 198 KEGG pathways, of which ‘endocytosis’ (ko04144), ‘MAPK signaling pathway’ (ko04010), ‘focal adhesion’ (ko04510), and ‘regulation of actin cytoskeleton’ (ko04810) annotated the most genes (Appendix A). Meanwhile, 20 KEGG pathways were significantly enriched (*q*-value < 0.05). The DEGs upregulated in ovaries and testes were involved in 17 and 12 significant pathways, respectively (Figure 4). Moreover, the enrichment analysis showed that clearly different pathways were enriched between the ovary- and testis-biased DEGs. The ovary-biased DEGs were significantly enriched in pathways such as ‘ribosome biogenesis in eukaryotes’, ‘DNA replication’, ‘pyrimidine metabolism’, ‘cell cycle’, and ‘spliceosome’ (Figure 4A), suggesting the importance of nucleic acid synthesis, protein homeostasis, and cell proliferation and differentiation for ovarian function. Whereas the testis-biased DEGs were highly enriched in pathways such as ‘focal adhesion’, ‘cytokine-cytokine receptor interaction’, ‘cell adhesion molecules (CAMs)’, ‘ECM (extracellular matrix)-receptor interaction’, and ‘MAPK signaling pathway’ (Figure 4B). Such noticeable differences imply the variation in the gene regulatory landscape between the *D. hystrix* ovary and testis.

### 3.5. Sex-Biased Genes of Interest Related to Reproduction Regulation

A number of GO terms and KEGG pathways known to be associated with gonadal development and maintenance, gametogenesis, oocyte maturation, and sperm motility were found in the present study, such as ‘reproductive process’ (GO:0022414), ‘gonad development’ (GO:0008406), ‘sexual reproduction’ (GO:0019953), ‘germ cell development’ (GO:0007281), ‘gamete generation’ (GO:0007276), ‘spermatogenesis’ (GO:0007283), ‘ovarian steroidogenesis’ (ko04913), ‘steroid hormone biosynthesis’ (ko00140), ‘Wnt signaling pathway’ (ko04310), and ‘MAPK signaling pathway’ (ko04010) (Appendix A). There were 221 and 618 genes grouped into reproduction-related GO terms and KEGG pathways, respectively (Appendix A). Based on the resources of annotation and enrichment analyses, 67 DEGs were obtained and these genes were determined to be involved in gonadal differentiation and development, gamete generation and maturation in vertebrates were identified, including gonadal soma-derived factor 1 (*gsdf1*), SRY-box transcription factor 9 (*sox9*), anti-Müllerian hormone (*amh*), doublesex- and mab-3-related transcription factor 1 (*dmrt1*), Cytochrome P450 aromatase (*cyp19a1a*), Kelch-like protein 10 (*klhl10*), zona pellucida sperm-binding protein 1 (*zp1*), and so on (Table 4). Among these DEGs, 45 were detected to be testis-biased, and 22 were ovary-biased. For example, the expressions of *amh* and *gsdf1* unigenes were upregulated in male gonads, whereas the expressions of zygote arrest protein 1 (*zar1*) and zona pellucida sperm-binding protein 3 (*zp3*) were found to be considerably higher in the ovaries (Table 4). In addition, we found that seven genes showed gonad-specific expression patterns. Of which, one gene (*cyp19a1a*) showed specific expression in ovaries; And the remaining six genes, such as *dmrt1*, steroid 11-beta-hydroxylase (*cyp11b1*), transcription factor SOX6 (*sox6*), and forkhead box protein J3 (*foxj3*) showed male-specific expression patterns. 

### 3.6. RT-qPCR Confirmation of DEGs

A total of 13 testis-upregulated and 10 ovary-upregulated DEGs were chosen and subjected to the statistical verification of expression profiles using RT-qPCR analysis. The relative expressions of these representative genes were shown in Figure 5. In general, the RT-qPCR results were found to be consistent with those of RNA-seq analysis (Figure 5). DEGs such as *amh*, *sox9*, *dmrt1*, and ropporin-1-like protein (*ropn1l*) were testis-biased (Figure 5A), whereas unigenes such as homologs of *zar1*, membrane-associated progesterone receptor component 1 (*pgrmc1*), and *vasa* were ovary-biased (Figure 5B). Meanwhile, a correlation analysis was conducted and the consistent tendencies of expression levels between the RNA-Seq data and RT-qPCR results (*R*^2^ = 0.8476) confirmed the reliability and accuracy of gene expression levels quantified by transcriptomic analysis (Figure 5C).

## 4. Discussion

Gonadal development from undifferentiated to differentiated stages and maturation is the most important determinant for the success of reproduction in fish. This highly complex biological process involves a set of functional genes that can promote the gonadal differentiation into either an ovary or a testis, and then cause a fish individual to exhibit a male or female phenotype [34]. To date, however, the molecular mechanisms underlying gonadal development have totally been unrevealed in *D. hystrix*. As an effective way to uncover the gene regulatory networks of gonad development and its dimorphism, transcriptome sequencing and comparative analysis between male and female gonads were employed to identify sex-related genes and to reveal their potential roles by combining differential expression data with biological pathways.

### 4.1. Sex-Biased Genes Involved in Steroids Synthetic Pathway

Sex steroid hormones, primarily including androgen and estrogen, influence the phenotypic sex by acting as key regulators for gonadal differentiation, development and sex maintenance in fish species [34,35]. In teleosts, the major androgen and estrogen essential to ovarian and testicular development are 11-ketotestosterone (11-KT) and 17β-estradiol (E_2_), respectively [36]. The syntheses of these sex steroid hormones need a series of genes encoding steroid-metabolizing enzymes. In particular, *cyp19a1a*, *cyp11b2* encoding steroid 11-beta-hydroxylase, *hsd11b1* coding for 11 beta-hydroxysteroid dehydrogenase, *cyp11a1* coding for cholesterol side-chain cleavage enzyme, and *hsd17bs* encoding 17β-hydroxysteroid dehydrogenases with 17-ketosteroid reducing activity are considered to be the most essential.

It has been demonstrated that *hsd11b1* and *cyp11b2* (encodes the key enzyme that converts testosterone to 11-KT) are involved in the pivotal steps in the synthesis of 11-KT in testis, whereas *cyp19a1a* gene product that catalyzes the conversion of androgens to estrogens is essential for the E_2_ synthesis in the ovary [37]. Previous reports have indicated that *cyp11b2* expression levels were comparatively higher in the testes of some teleost fishes [38,39], implicating a regulatory role for *cyp11b2* gene in testicular development. In this study, *cyp11b2* gene was expressed at significantly higher levels in *D. hystrix* testes, while *cyp19a1a* was found to be upregulated in ovaries compared to testes, exhibiting a similar expression pattern to other fish species [34,37,40,41]. The findings suggested that these genes play potential roles in the development of gonads and may participate in *D. hystrix* reproduction. Interestingly, the sexual dimorphism of serum E_2_ and 11-KT levels has already been observed in fish species; the levels of E_2_ and 11-KT are relatively higher in females and males, respectively [34,41]. In *Oreochromis niloticus*, both serum E_2_ levels and *cyp19a1a* expression were comparatively higher in females than in males [41]. In *Oryzias latipes*, mutation of *cyp19a1a* gene led to a marked decrease in the gonadal E_2_ level in female (XX) individuals [42]. Collectively, higher serum E_2_ levels may be attributed to the elevated ovarian *cyp19a1a* expression.

Steroidogenic acute regulatory protein (StAR) participates in the rate-limiting step of steroid biosynthesis by transporting cholesterol to the inner mitochondrial membrane where the cholesterol side-chain cleavage enzyme converts this substrate to pregnenolone [34,35]. In teleosts, the *star* gene was found to be highly expressed during spermatogenesis, oocyte maturation and ovulation [43,44]. Our study showed that the expression levels of *star* and *cyp11a1* in the testes of *D. hystrix* were significantly higher than those in the ovaries. The current results showed good agreement with the experimental results in *Scatophagus argus* [34] and *P. olivaceus* [45], suggesting that the abundances of *star* and *cyp11a1* may partially affect the rate of steroid synthesis in teleost fish. In fish gonads, sex steroids synthesis-related genes are modulated by the hypothalamic–pituitary–gonadal (HPG) axis, one of the key steps is cAMP-mediated stimulation of *star* expression [34].

In *O. niloticus*, *star* mRNA levels in testes were greatly enhanced by injection of human chorionic gonadotropin (hCG) [46]. Moreover, recombinant follicle stimulating hormone (FSH)/luteinizing hormone (LH) administration increased the *star* and *hsd11b1* expressions, E_2_/11-KT levels, and finally promoted the ovary and testicular development in *S. argus* [47]. Thus, understanding the expression and endocrine regulation of steroidogenic genes would greatly help us establish effective methods for controlling reproduction in *D. hystrix* aquaculture, such as multiple gonadotropin-releasing hormone agonist (GnRHa) injections or implants that are commonly utilized for the artificial induction of oocyte maturation/ovulation and spermination in fish.

### 4.2. Candidate Genes Related to Gonad Differentiation and Development

The molecular mechanisms involved in sex determination and gonad differentiation are variable among phyla. Although the top upstream regulators in the sex determination pathway are less conserved, the downstream genes are more conserved. With rare exceptions, almost all currently identified sex-determining genes belong to one of the three protein families (Dmrt, TGF-β and its signaling pathway, and Sox) [21].

Here, regulatory genes that appear to be involved in male gonad differentiation were identified from the *D. hystrix* gonad transcriptomes. In particular, detection of highly expressed *dmrt1* as a male-biased gene would be of great interest. *Dmrt1* belonging to the Dmrt gene family generally functions as a conserved transcription factor in the sexual regulatory cascade. *Dmrt1* and its paralogs have been claimed as master sex-determininggenes in some animal species [21,34], playing essential parts in the differentiation of testis and maintenance of male-specified germ cells [48]. Also, it has been understood that *dmrt1* works as an essential factor in gonadal development and gametogenesis in fishes [20,49]. Dmrt genes stimulate male-specific differentiation but repress female-specific differentiation [21]. In *O. latipes*, the mutation of autosomal *dmrt1* was found to be responsible for a male-to-female sex reversal [50]. Comparably, *dmrt1*-mutated testes exhibited serious testicular development defects and gradual loss of germ cells in zebrafish [51]. In this research, *dmrt1* genes were determined to be specifically expressed in the testis; ovarian *dmrt1* expression could not be detected by transcriptome analysis and RT-qPCR. The trend of *dmrt1* expression was quite similar to those in other fishes such as *O. niloticus*, and *Megalobrama amblycephala* [40,52], suggesting that *dmrt1* gene is a key player in the testis development in *D. hystrix*.

A series of sex-determining genes encoding transforming growth factor-β (TGF-β) signal components (e.g., *Gsdf^Y^*, *amhy*, *Amhr2*, *Gdf6Y*) have been identified in fish, suggesting that the TGF-β pathway is involved in gonad differentiation. Gonadal soma-derived factor (Gsdf), a TGF-β superfamily member, is found to be expressed specifically in fish gonads, predominantly in the Sertoli cells and neighboring spermatogonia of testes [53]. As a direct downstream gene of *dmrt1*, Jiang et al. found that *gsdf* gene transcription was regulated by *dmrt1* [53]. Recently, the authors further demonstrated that *dmrt1* could induce the expression of *gsdf* with the participation of splicing factor 1 (SF-1, also known as Nr5a1, an important activator of steroidogenic enzymes, including aromatase) [54]. 

Previous studies have shown that *gsdf* plays a key role in testicular differentiation in fish, and it is speculated that *gsdf* acts by suppressing the activator of *cyp19a1a* and inhibiting estrogen synthesis [53]. Mutation of *gsdf* in medaka and *O. niloticus* initiated male-to-female sex reversal [53,55], while overexpression of this gene induced testis differentiation in female *O. niloticus* [56]. A study involving *Oncorhynchus mykiss* showed that *gsdf* may act in the regulation of spermatogenesis by stimulating the proliferation of spermatogonia [57]. In teleost, it was reported that *gsdf* was expressed at a higher level in the testicular somatic cells compared with ovarian tissues [58]. *Sf-1* was significantly upregulated during and after testicular differentiation in black porgy [59]. Similar trends of *gsdf* and *sf-1* expressions were also observed in this study. Therefore, we could deduce that *gsdf* has a conserved function in the testis differentiation of *D. hystrix*.

Anti-Müllerian hormone (Amh) encoded by *amh* has also been identified as a member of the TGF-β family in fish species [18]. *Amh* suppresses the development of the Müllerian ducts and functions as a key regulator for differentiation of the Sertoli and granulosa cells, germ cell proliferation and steroidogenesis in Leydig cells in gonad development [34]. Lin et al. [51] found that *amh* mutation resulted in a female-biased sex ratio in zebrafish; the unrestrained germ cell proliferation in male *amh* mutants led to hypertrophic testes. In XY medaka, Amh type II receptor (*amhr2*) mutation could promote the sex reversal and *amhr2* mutants mostly exhibited the signs of germ cell over-proliferation [60]. Our data showed that the expressions of *amh* and *amhr2* genes were upregulated in the testes but weakly expressed in the ovaries, implicating the significance of Amh/Amhr2 pathway in the modulation of testicular differentiation and germ cell proliferation in *D. hystrix*.

Several members of the Sox (SRY-related HMG box) gene family has also been found to regulate the differentiation of gonads in fish; typical examples include *sox9*, *sox8*, *sox5*, and *sox3* [18,61]. Here, the abundances of the two transcriptional factors *sox9* and *sox6* were detected in our transcriptome data and they were identified as male-biased genes. Classic studies have clearly demonstrated that *sox9* plays vital roles in the testicular development of male gonad as an important sex-determination gene [35]. *Sox9* was found to be expressed in the testes of rainbow trout [62], and channel catfish [63]. Its crucial role in sex determination of teleost fish has also been confirmed by genetic approaches [21]. Genomic studies have revealed that the *sox9* gene in teleosts has undergone duplication and there are two copies (*sox9a* and *sox9b*) [34,61]. In both male and female medaka, *sox9b* was shown to be pivotal for the survival of germ cells [64]. Certain regulatory genes in male fish may regulate the expression of *sox9b* mRNA in teleost fish. A recent study demonstrated that the Nile tilapia *dmrt1* gene positively regulated the transcription of *sox9b* by directly binding to a specific promoter cis-regulatory element [61]. Moreover, *sox6* gene was reported to be specifically expressed in the testis and involved in the later stages of spermatogenesis in teleost fishes [65]. In this study, both RNA-seq and RT-qPCR analysis demonstrated that *sox9* and *sox6* mRNA levels were much higher in the male gonads. Such expression patterns were similar to those observed in *O. latipes* [66], *S. Sillago* [18], and *S. argus* [34], suggesting a highly conserved function of *sox9* gene in teleosts.

As to the identification of genes that participate in the differentiation of gonads, it is necessary to explore their roles in both undifferentiated and differentiated gonad samples. Importantly, detailed information about the gonad differentiation-related genes should be collected and explored in more developmental stages, resembling what has been performed in other fish species [34]. Hence, the expression profiles of the sex-biased genes identified in this study should be carefully analyzed in the future. In this study, the sex-specific expression of *dmrt1* implicates it as the most promising candidate sex-determining gene in *D. hystrix*. In order to further confirm the precise role of *dmrt1*, molecular genetic studies are firstly required to characterize the transcriptional regulation of putative downstream genes (e.g., *gsdf*, and *sf-1*). Meanwhile, the DMRT gene cluster will be cloned, and then gene structures will be comparatively analyzed, to identify any possible sex-linked polymorphic locus. With the foundation of above basic research, specific gene knockout and rescue experiments would be required to the functional confirmation of *dmrt1* gene in sex determination.

### 4.3. Sex-Biased Genes Involved in Gametogenesis and Gamete Maturation

In fish aquaculture, the eventual aim of artificially induced breeding is to facilitate the development of gonads and then obtain mature gametes [26]. In this study, the expression of many genes associated with oogenesis, oocytes maturation, spermatogenesis and sperm motility was presented, such as *vasa*, zygote arrest 1 (*zar1*), zona pellucida sperm-binding proteins (*zps*), spermatogenesis-associated proteins (*spatas*), and spermine oxidase (*smox*). Vasa, specifically localized in germline cells, has been well characterized as a crucial player in germ cell formation in larval fish [35]. A previous study has pointed out that *vasa* gene is expressed in primordial germ cells during the first formation of the gonadal anlagen [67]. In Pacific bluefin tuna (*Thunnus orientalis*), *vasa* exhibited a high level of expression in spermatogonia, as well as oogonia and previtellogenic oocytes [68]. The *D. hystrix* transcriptomic data showed that *vasa* mRNA was present both in the male and female gonads, with a drastically higher mRNA level in the ovaries. Such result is consistent with those reported in southern bluefin tuna (*Thunnus maccoyii*) [69] and yellow catfish (*Pelteobagrus fulvidraco*) [70], implying that *vasa* play a certain role in female gonad development. However, the expression pattern of *vasa* shows large variations among different species [35]. The functional details of *vasa* in gonadal differentiation in *D. hystrix* need to be clarified in future studies.

In vertebrates, *zar1* acts as an oocyte-specific maternal effect gene with an evolutionarily conserved function in the fertilization process [34,35]. The zona pellucida (ZP) encoded by *zp* genes is a glycoproteinaceous matrix surrounding the oocyte and has an important part in species-specific binding of sperm [18]. In zebrafish, the *zp* mRNA levels were enhanced notably during oogenesis, particularly at the previtellogenic stage [71]. It has been shown that *zp1* participates in the generation of oocyte envelope, and *zp3* acts as a major class of female-specific factors in the reproductive process [72]. In the present study, *zp1* and *zp3* were more highly expressed in the ovaries than in the testes, suggesting that these oocyte-specific genes appear to be involved in ovarian folliculogenesis in *D. hystrix*. Regarding male reproduction, sperm acrosome membrane-associated protein 6 (*spaca6*) can mediate the fusion between ovum and spermatozoa as a sperm membrane component [73]. It has been demonstrated that *spatas* and sperm surface protein Sp17 (*sp17*) provide vital functions for spermatogenesis and sperm motility [74]. Here, significantly higher expressions of genes *spaca6*, *spata5*, *spata6*, *spata7*, *smox* and *sp17* were observed in the male gonad; the expression difference may be associated with the spermatogenesis in *D. hystrix*. Absolutely, these identified gonocyte-specific genes would be helpful for investigating the control mechanisms during oogenesis and spermatogenesis in *D. hystrix*. In order to guarantee the performance of reproductive management of this fish species under aquaculture conditions, more importantly, further efforts to establish an effective technique for inducing gamete maturation are highly encouraged.

## 5. Conclusions

This is the first study on the gonad transcriptome of *D. hystrix* and a number of 57,077 unigenes were assembled. A comparison of ovarian and testicular transcriptomes detected a set of DEGs supposedly involved in gonadal development and gametogenesis. The conserved expression profiles of the well-known reproduction-related genes imply their similar roles in the gonad differentiation and development of *D. hystrix*. Our findings offer a precious data source for further investigation into the regulatory mechanisms and molecular characteristics of the reproductive process in *D. hystrix*.

## Figures and Tables

**Figure 1 animals-11-01042-f001:**
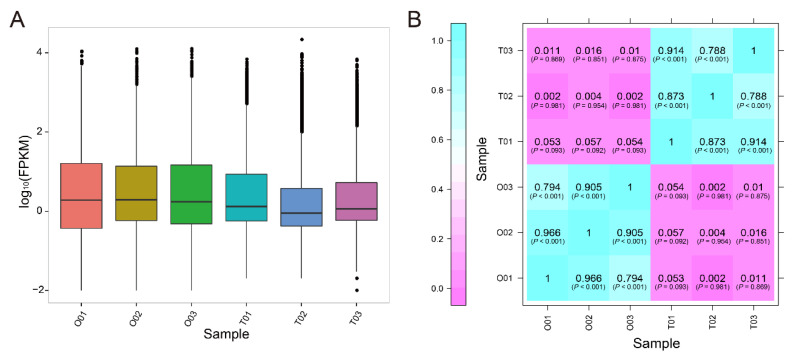
Gene expression pattern analysis in gonads of *D. hystrix*. (**A**) The distribution of gene expression levels. The horizontal axis indicates sample name, whereas the vertical axis indicates log_10_ (FPKM) value. Box plots indicate the maximum, top quartile, median, bottom quartile, and minimum values from top to bottom. (**B**) Heatmap representation of the correlations between each two samples. The color represents the correlation coefficient. O, ovary; T, testis.

**Figure 2 animals-11-01042-f002:**
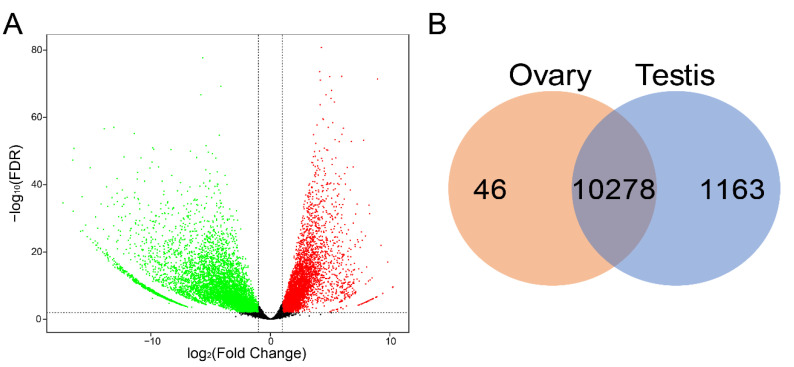
Differential gene expression between the ovaries and testes of *D. hystrix*. (**A**) Volcano plot of the differences in gene expression. Red dots: upregulated, represent ovary-biased genes; Green dots: downregulated, represent testis-biased genes. (**B**) Venn diagram showing the distribution of testis-specific and ovary-specific DEGs.

**Figure 3 animals-11-01042-f003:**
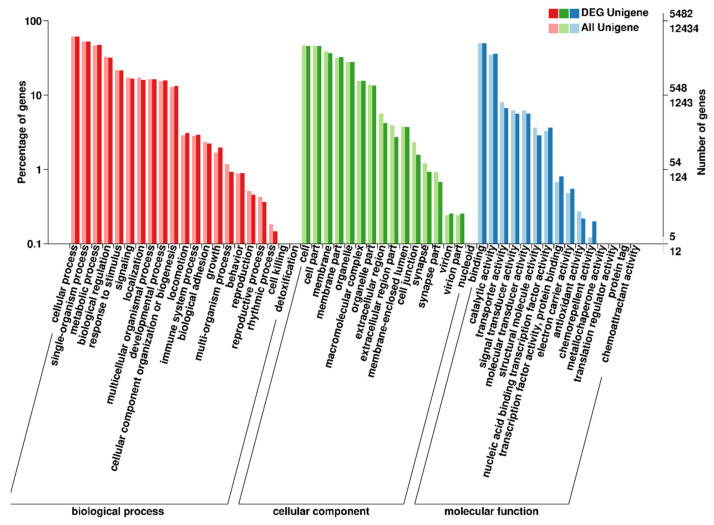
Functional annotation of DEGs based on GO categorization. The horizontal axis indicates the GO functions, and the vertical axis represents the number of genes with GO function.

**Figure 4 animals-11-01042-f004:**
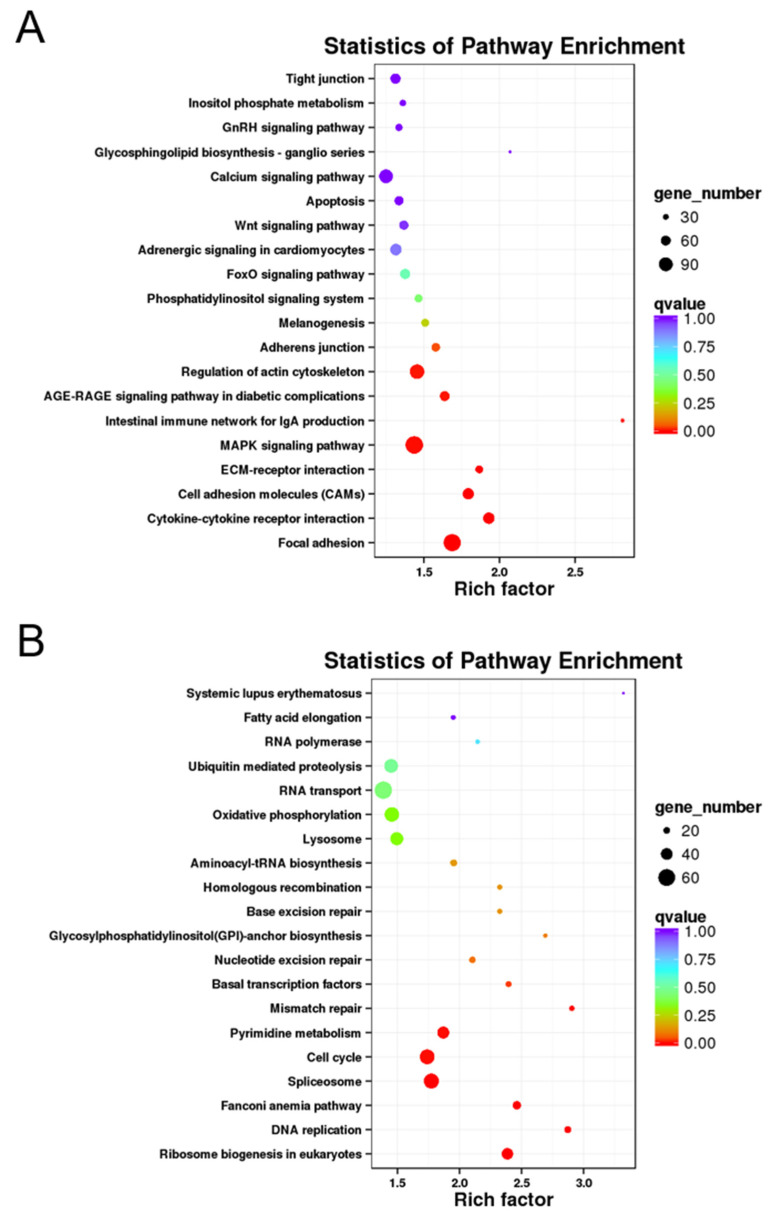
Significantly enriched KEGG pathways of DEGs. (**A**) Pathway enrichment analysis of DEGs upregulated in testes. (**B**) Pathway enrichment analysis of DEGs upregulated in ovaries. The pathways and rich factor are shown in the vertical and the horizontal axis, respectively. The size of the point indicates the number of genes, and the color indicates the *q* value.

**Figure 5 animals-11-01042-f005:**
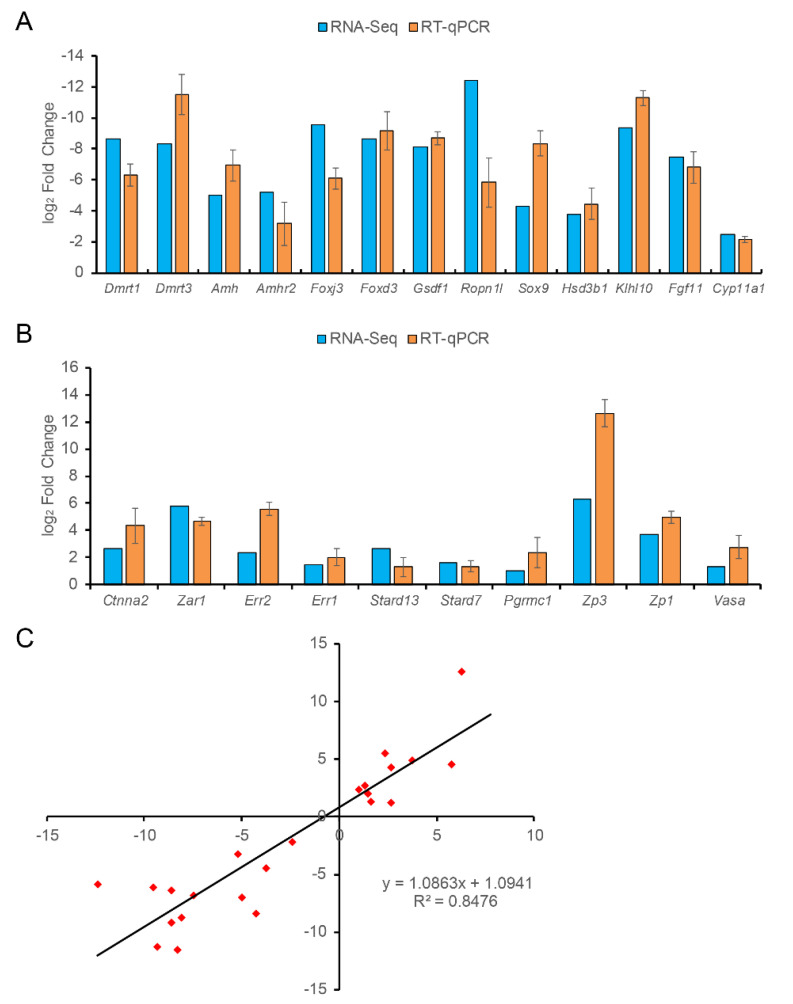
Verification of expression profiles of 13 testis-biased (**A**) and 10 ovary-biased genes (**B**) using RT-qPCR. (**C**) Correlation analysis of the RNA-Seq data and RT-qPCR results.

**Table 1 animals-11-01042-t001:** Summary statistics of the gonadal RNA-Seq data for *D. hystrix*.

Sample	Raw Reads	Clean Reads	Clean Bases (bp)	GC Content (%)	% ≥ Q20	% ≥ Q30
Ovary 1	23,164,033	22,312,200	6,686,832,318	48.66	98.06	94.32
Ovary 2	22,001,854	21,207,110	6,354,910,044	48.92	97.53	93.24
Ovary 3	21,490,133	20,679,487	6,198,085,778	48.86	97.47	93.14
Testis 1	23,845,599	23,454,828	7,010,039,930	48.74	98.32	95.25
Testis 2	40,744,744	39,306,614	11,739,627,328	48.52	98.31	95.16
Testis 3	25,338,572	24,928,306	7,457,433,494	48.57	98.17	94.83
Mean	26,097,489	25,314,758	7,574,488,149	48.71	97.98	94.32
Total	156,584,935	151,888,545	45,446,928,892			

**Table 2 animals-11-01042-t002:** Summary statistics of the *D. hystrix* gonadal transcriptome assembly.

Length Range	Transcript	Unigene
300–500 bp	30,973 (22.18%)	23,402 (41.00%)
500–1000 bp	30,900 (22.13%)	14,055 (24.62%)
1000–2000 bp	31,636 (22.66%)	8495 (14.88%)
>2000 bp	46,119 (33.03%)	11,125 (19.49%)
Total number	139,628	57,077
Total length (bp)	261,271,796	74,197,236
N50 length (bp)	3118	2560
Mean length (bp)	1871.2	1299.9

**Table 3 animals-11-01042-t003:** Statistics of the *D. hystrix* gonadal transcriptome annotation.

Database	Annotated Number	Percentage	300 ≤ Length < 1000 bp	Length ≥ 1000 bp
COG	6281	11.00%	1522	4759
GO	12,434	21.78%	3570	8864
KEGG	13,699	24.00%	3705	9994
KOG	15,157	26.56%	4048	11,109
Pfam	17,519	30.69%	4206	13,313
Swiss-prot	13,488	23.63%	2933	10,555
eggNOG	22,671	39.72%	7438	15,233
Nr	23,114	40.50%	7469	15,645
All	24,574	43.05%	8824	15,750

**Table 4 animals-11-01042-t004:** Searching for differential expression genes (DEGs) putatively involved in reproduction from the gonad transcriptome of *D. hystrix*.

Unigene ID	Gene Annotation	Log_2_ Fold Change (Ovary/Testis)	FDR ^1^
Relatively higher expression in testis			
c84183.graph_c0	Gonadotropin-releasing hormone II receptor	−3.63	1.23E-06
c81599.graph_c0	Doublesex- and mab-3-related transcription factor 1	−8.64	5.85E-07
c87605.graph_c0	Doublesex- and mab-3-related transcription factor 3	−8.34	4.17E-51
c69078.graph_c0	Anti-Mullerian hormone	−5.03	1.14E-11
c89331.graph_c1	Anti-Muellerian hormone type-2 receptor	−5.19	1.37E-20
c90445.graph_c0	Forkhead box L3	−7.13	7.21E-29
c83812.graph_c1	Forkhead box protein J3	−9.57	1.21E-08
c80643.graph_c0	Forkhead box protein D3	−8.62	7.63E-07
c90293.graph_c0	Wilms tumor protein 1-interacting protein	−2.73	6.19E-14
c86435.graph_c1	Wilms tumor protein homolog	−3.11	4.20E-07
c87662.graph_c6	Wilms tumor protein	−2.75	1.14E-06
c74886.graph_c0	Gonadal soma derived factor 1	−8.11	8.58E-34
c67874.graph_c1	Transcription factor SOX9	−4.29	1.13E-07
c73370.graph_c0	Transcription factor SOX6	−9.46	2.54E-08
c80342.graph_c0	Steroidogenic acute regulatory protein	−5.22	1.05E-26
c88056.graph_c0	17-beta-hydroxysteroid dehydrogenase 14	−3.92	1.12E-17
c74550.graph_c0	3-oxo-5-beta-steroid 4-dehydrogenase	−2.82	1.41E-03
c56757.graph_c0	3 beta-hydroxysteroid dehydrogenase type 1	−3.75	9.81E-10
c79716.graph_c0	Steroid 11-beta-hydroxylase	−8.40	1.94E-11
c83337.graph_c1	StAR-related lipid transfer protein 13	−5.63	2.00E-19
c89692.graph_c0	StAR-related lipid transfer protein 9	−1.59	2.13E-05
c80342.graph_c0	Steroidogenic acute regulatory protein	−5.22	1.05E-26
c89905.graph_c0	Estrogen receptor b2	−2.41	2.43E-09
c84046.graph_c0	Estrogen receptor	−2.86	1.55E-10
c82367.graph_c0	Estrogen receptor b1	−3.06	5.18E-13
c89150.graph_c2	Progesterone receptor	−1.34	6.89E-04
c86613.graph_c1	Kelch-like protein 10	−9.33	3.60E-31
c85739.graph_c0	Spermatogenesis-associated protein 5	−1.95	1.19E-04
c87439.graph_c0	Spermatogenesis-associated protein 20	−2.16	4.59E-04
c86517.graph_c0	Spermatogenesis-associated protein 7	−4.85	4.25E-09
c83923.graph_c0	Spermatogenesis-associated protein 17	−6.61	1.43E-13
c84036.graph_c0	Spermatogenesis-associated protein 6	−3.16	5.66E-10
c81488.graph_c0	Sperm surface protein Sp17	−1.77	2.49E-03
c82755.graph_c0	Sperm acrosome membrane-associated protein 6	−3.09	2.95E-05
c42266.graph_c0	Spermine oxidase	−5.52	7.33E-09
c75610.graph_c0	Ropporin-1-like protein	−12.40	3.92E-39
c86581.graph_c5	Splicing factor 1	−2.21	8.72E-04
c87182.graph_c0	Fibroblast growth factor receptor-like 1	−6.42	4.76E-36
c86547.graph_c1	Fibroblast growth factor 11	−7.51	2.39E-10
c73147.graph_c0	Fibroblast growth factor 7	−3.52	5.76E-05
c77489.graph_c0	Fibroblast growth factor-binding protein 1	−3.12	2.17E-05
c81519.graph_c1	Fibroblast growth factor 10	−6.61	1.31E-11
c84573.graph_c2	Fibroblast growth factor-binding protein 3	−2.05	7.40E-03
c89383.graph_c0	Fibroblast growth factor receptor 2	−5.21	1.75E-13
c63200.graph_c0	ATP synthase F0 subunit 6	−2.86	6.13E-33
c78426.graph_c0	Cholesterol side-chain cleavage enzyme	−2.44	7.91E-05
Relatively higher expression in ovary			
c73686.graph_c0	Gonadotropin subunit beta-2	4.47	1.46E-28
c80357.graph_c1	Catenin beta-1	1.79	3.63E-07
c85139.graph_c8	Catenin alpha-2	2.64	7.20E-23
c80092.graph_c0	Protein fem-1 homolog C	2.88	8.07E-13
c85792.graph_c0	Protein fem-1 homolog B	2.26	7.47E-10
c83767.graph_c0	Zygote arrest protein 1	5.76	2.60E-14
c82641.graph_c1	3-oxo-5-alpha-steroid 4-dehydrogenase 1	1.40	1.13E-03
c88743.graph_c5	Steroid hormone receptor ERR2	2.34	4.22E-12
c87163.graph_c3	3-keto-steroid reductase	1.90	2.24E-05
c73009.graph_c0	Inactive hydroxysteroid dehydrogenase-like protein 1	1.11	3.55E-04
c85481.graph_c3	Steroid hormone receptor ERR1	1.47	8.53E-04
c73497.graph_c0	Hydroxysteroid dehydrogenase-like protein 2	1.14	3.41E-04
c72251.graph_c0	Cytochrome P450 aromatase	3.86	8.36E-08
c87866.graph_c0	StAR-related lipid transfer protein 13	2.63	1.17E-10
c83997.graph_c0	StAR-related lipid transfer protein 7	1.63	4.78E-07
c82351.graph_c1	Membrane-associated progesterone receptor component 1	1.01	2.73E-03
c83997.graph_c2	Membrane-associated progesterone receptor component 2	2.87	6.61E-09
c83333.graph_c2	Progesterone-induced-blocking factor 1	1.66	4.04E-08
c42601.graph_c0	Zona pellucida sperm-binding protein 3	6.29	2.60E-07
c85704.graph_c1	Zona pellucida sperm-binding protein 1	3.72	2.40E-29
c87366.graph_c0	Vasa	1.33	1.15E-04
c78173.graph_c1	Forkhead box protein H1	4.80	6.46E-41

^1^ False discovery rate.

## Data Availability

The data presented in this study are available on request from the corresponding author. The data are not publicly available due to the agreement with funding bodies.

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
