# Peer review of "Comparison of Gonadal Transcriptomes Uncovers Reproduction-Related Genes with Sexually Dimorphic Expression Patterns in *Diodon hystrix"

_animals, 2021, doi:10.3390/ani11041042_

Round 1

Reviewer 1 Report

General comments

First of all congratulations to the authors for an extensive transcriptome analysis of a new species. The authors are presenting the first attempt of gonadal transcriptome analysis of D. hystrix. The results of their analyses are presented in a suitable way. The methodology followed is of standard analytical methods. However, the use of English of the manuscript vastly needs improvements. Moreover, there are comments in the discussion not agreeing with the results presented.

The introduction is focused on aquaculture species and the importance of a decently tuned reproductive management of an aquaculture species (sometimes not suitable terminology is used for that). However, there are no examples or reference presented for the aquaculture of the species under study. Is it a fish of high commercial value that could make potentially a profitable aquaculture operation? It seems that this is the case since it is mentioned that “The consumer fish still has to be captured from the wild”. Moreover, there is no information concerning the type of sex determination of the species (e.g. is it protandrous hermaphrodite or protogynous hermaphrodite? etc.) the reader needs to reach the discussion in order to realize that is a sexually dimorphic species. Nevertheless, if the species is not used in aquaculture there are other areas apart from the reproductive management that have to be addressed. I think most of the introduction should be changed in terms of structure and focus on the commercial importance of the species and on how the results of the study could assist towards such a plausible scenario (i.e. aquaculture of this species).  Furthermore, it needs even a small part of the discussion to be dedicated to how the results could be utilized in the management of reproduction of the species under captivity, since this the main reason that initiated the research interest, as the authors are stating in the introduction. I think the results of the RT-PCR study could provide the basis for such discussion.

Specific comments

Pg 1 l 21-22 “… artificial propagation remains a major technological barrier for sustainable development of the porcupine fish industry.” change to “… management of breeding and reproduction under captivity remains a major technological barrier for sustainable development of the porcupine fish industry.”

Pg 1 l 30-31 “… artificial propagation remains a major technological barrier for sustainable development of the porcupine fish industry.” change to “… management of breeding and reproduction under captivity remains a major technological barrier for sustainable development of the porcupine fish industry.”

Pg 2 l 54-55 “The consumer fish still has to be captured from the wild.” This means that the fish is not farmed yet (i.e. there is no aquaculture for it). But in the abstract the authors mention that is “a new emerging aquaculture species”. How it is emerging when the authors do not reference any efforts or examples of even experimental aquaculture of the species. If this is the case it should be mentioned in the manuscript as potentially aquaculture species. Moreover, the sentence should be changed to “The consumer’s fish still has to be captured from the wild.”, or better restructured.

Pg 2 l 58-59 Again the authors mention “… sustainable aquaculture of….”. It has to exist in order to be sustainable. Previously the authors mention that “The consumer fish still has to be captured from the wild.” Which of the two cases is true a) is there any aquaculture with difficulties or b) there is a demand for the development of aquaculture for this species because of fish resource has been declining etc.? Moreover, the sentence should be restructured “Thus, the sustainable aquaculture of D. hystrix is absolutely in great and urgent need of artificial breeding and reproduction management.”

Pg 2 l 58-59 “…the development of a perfect artificial reproduction technology,…” the word “perfect” I think is not suitable, better change to “…the development of an efficient management of reproduction,…”.

Pg 2 l 71-75. The sentence is too long and with no correct use of English. Please rephrase.

Pg 2 l 76-77. “Compared with other economic fishes, however, the genetic information available for D. hystrix is rather limited.” change to “Compared with other commercially aquaculture species, however, the genetic information available for D. hystrix is rather limited.”.

Pg 2 l 80-83. Please rewrite the sentence. Transcriptome sequencing is not genetic information since it is related to RNA and not DNA. RNA and its expression can be influenced by non genetic factors.

Pg 2 l 93-94. “These studies provide helpful insights into the reproduction related genes and enable the discovery of new gene candidates.” change to “These studies provide helpful insights into reproduction related genes and enable the discovery of new gene candidates.”

Pg 3 l 98-99. “… the expression differences of the genes potentially involved in reproduction regulation 98 were analyzed and discussed.” change to “… the differences in the expression of the genes which potentially are involved in regulation of reproduction were analyzed and discussed.”

Pg 3 l 117-119. “The testis and ovary tissues were excised from D. hystrix individuals as soon as possible and quick-frozen in liquid nitrogen immediately, and then stored at -80°C until RNA extraction.” Change to “The testis and ovary tissues were excised from D. hystrix individuals within … from sacrifice, immediately quick-frozen in liquid nitrogen, and then stored at -80°C until RNA extraction.”

Pg 3 l 117-119 ”… end (PE) reads with a length of 125 bp were generated.” Change to “… end (PE) reads of 125 bp length were generated.”

Pg 3 l 136-137 “By means of SOAPnuke (version 1.5.0) [24], the raw sequencing data were quality- controlled with the parameters “-l 10 -q 0.5 -n 0.05 -p 1 -i”.” change to “By means of SOAPnuke (version 1.5.0) [24], the raw sequencing data were pruned using the software’s quality control with the parameters “-l 10 -q 0.5 -n 0.05 -p 1 -i”.”

Pg 6 l 228-229 “… or not to be presented (0 ≤ FPKMs < 0.1)…” change to “… or not expressed at all (0 ≤ FPKMs < 0.1)…”

Pg 13 l 395-397 “Thus, understanding the expression and endocrine regulation of steroidogenic genes would greatly help us establish better methods for controlling reproduction in D. hystrix aquaculture.” How it would help us in a practical aquaculture breeding terms? I understand that further research is needed to reach that level but an example of this kind of further research should be given.

Pg 13 l 416-418 “In the present study, the transcript levels of dmrt1 were determined to be significantly higher in the male gonads than in the female gonads, ovarian dmrt1 expression could not be detected by transcriptome analysis (Table 4).” It is confusing this sentence when is compared to the results.

For example:

In figure 5 with RT-PCR you present expression of dmrt1 but not in the ovary. From what it can be seen in the results, there is no significantly higher expression difference in male and female. In male is expressed but in females is not according to the RT-PCR.

 In the transcriptome analysis you mention that there is no expression detected. However in the results you mentions as DGEs the “doublesex- and mab-3-related transcription factor 1 (dmrt1)” and present it in Table 4.

Pg 13 l 478-481 “Hence, the expression profiles of the sex-biased genes identified in this study should be carefully analyzed in the future, and further molecular biology and genetic studies are required to clarify the sexual manipulation roles of these genes, and then confirm the sex determining gene.” A correct statement but it can be supported from a hypothetical example coming out from the results of this study could be given and discussed.

Pg 15 l 484-486 “In this study, many genes associated with oogenesis, oocytes maturation, spermatogenesis and sperm motility were recognized, such as vasa,...” change to... “In this study, the expression of many genes associated with oogenesis, oocytes maturation, spermatogenesis and sperm motility was presented, such as vasa,...”.

Pg 16 l 517-518 “In order to guarantee the performance of artificial propagation, more…” change to “In order to guarantee the performance reproductive management of this fish species under aquaculture conditions, more…”.

Author Response

General comments: First of all congratulations to the authors for an extensive transcriptome analysis of a new species. The authors are presenting the first attempt of gonadal transcriptome analysis of D. hystrix. The results of their analyses are presented in a suitable way. The methodology followed is of standard analytical methods. However, the use of English of the manuscript vastly needs improvements. Moreover, there are comments in the discussion not agreeing with the results presented.

The introduction is focused on aquaculture species and the importance of a decently tuned reproductive management of an aquaculture species (sometimes not suitable terminology is used for that). However, there are no examples or reference presented for the aquaculture of the species under study. Is it a fish of high commercial value that could make potentially a profitable aquaculture operation? It seems that this is the case since it is mentioned that “The consumer fish still has to be captured from the wild”. Moreover, there is no information concerning the type of sex determination of the species (e.g. is it protandrous hermaphrodite or protogynous hermaphrodite? etc.) the reader needs to reach the discussion in order to realize that is a sexually dimorphic species. Nevertheless, if the species is not used in aquaculture there are other areas apart from the reproductive management that have to be addressed. I think most of the introduction should be changed in terms of structure and focus on the commercial importance of the species and on how the results of the study could assist towards such a plausible scenario (i.e. aquaculture of this species).  Furthermore, it needs even a small part of the discussion to be dedicated to how the results could be utilized in the management of reproduction of the species under captivity, since this the main reason that initiated the research interest, as the authors are stating in the introduction. I think the results of the RT-PCR study could provide the basis for such discussion.

Reply: Thank you much for the valuable comments. As you pointed out, Diodon hystrix is just a promising species for commercial aquaculture at present. In the light of your suggestions, the introduction has been re-structured and some pertinent points has been addressed additionally. The information concerning the type of sex determination of the species has also been added. Moreover, we have improved the discussion section as you suggested, and the language has been checked throughout the revised manuscript by an English native speaker.

Specific comments

Comment 1: Pg 1 l 21-22 “… artificial propagation remains a major technological barrier for sustainable development of the porcupine fish industry.” change to “… management of breeding and reproduction under captivity remains a major technological barrier for sustainable development of the porcupine fish industry.”

Reply: Revised as requested(line 21-23).

Comment 2: Pg 1 l 30-31 “… artificial propagation remains a major technological barrier for sustainable development of the porcupine fish industry.” change to “… management of breeding and reproduction under captivity remains a major technological barrier for sustainable development of the porcupine fish industry.”

Reply: Revised as requested(line31-32).

Comment 3: Pg 2 l 54-55 “The consumer fish still has to be captured from the wild.” This means that the fish is not farmed yet (i.e. there is no aquaculture for it). But in the abstract the authors mention that is “a new emerging aquaculture species”. How it is emerging when the authors do not reference any efforts or examples of even experimental aquaculture of the species. If this is the case it should be mentioned in the manuscript as potentially aquaculture species. Moreover, the sentence should be changed to “The consumer’s fish still has to be captured from the wild.”, or better restructured.

Reply: We agree with your viewpoint and apologize for the inaccurate description here. Diodon hystrix is indeed a promising species for commercial aquaculture. The sentence ‘The consumer fish still has to be captured from the wild’ has been changed to ‘The commodity fish still has to be captured from the sea’ (line 65).

Comment 4: Pg 2 l 58-59 Again the authors mention “… sustainable aquaculture of….”. It has to exist in order to be sustainable. Previously the authors mention that “The consumer fish still has to be captured from the wild.” Which of the two cases is true a) is there any aquaculture with difficulties or b) there is a demand for the development of aquaculture for this species because of fish resource has been declining etc.? Moreover, the sentence should be restructured “Thus, the sustainable aquaculture of D. hystrix is absolutely in great and urgent need of artificial breeding and reproduction management.”

Reply: As we already explained in the previous replies for general comments and comment 3, Diodon hystrix is just a promising species for commercial aquaculture. The fact of the matter is that the population of D. hystrix has been declining in these years as a result of a sharp increase of fish catch and a weak resilience of its natural population. This sentence has been restructured to ‘Thus, the large-scale aquaculture of D. hystrix is absolutely in great and urgent need of artificial breeding and reproduction management’(line69-70). Thanks.

Comment 5: Pg 2 l 58-59 “…the development of a perfect artificial reproduction technology,…” the word “perfect” I think is not suitable, better change to “…the development of an efficient management of reproduction,…”.

Reply: Revised as requested(line 75-78).

Comment 6: Pg 2 l 71-75. The sentence is too long and with no correct use of English. Please rephrase.

Reply: This sentence has been rephrased as requested(line84-90).

Comment 7: Pg 2 l 76-77. “Compared with other economic fishes, however, the genetic information available for D. hystrix is rather limited.” change to “Compared with other commercially aquaculture species, however, the genetic information available for D. hystrix is rather limited.”

Reply: Revised as requested(line91-92).

Comment 8: Pg 2 l 80-83. Please rewrite the sentence. Transcriptome sequencing is not genetic information since it is related to RNA and not DNA. RNA and its expression can be influenced by non genetic factors.

Reply: This sentence has been rewritten(line95-100).

Comment 9: Pg 2 l 93-94. “These studies provide helpful insights into the reproduction related genes and enable the discovery of new gene candidates.” change to “These studies provide helpful insights into reproduction related genes and enable the discovery of new gene candidates.”

Reply: Revised as requested(line110-111).

Comment 10: Pg 3 l 98-99. “… the expression differences of the genes potentially involved in reproduction regulation 98 were analyzed and discussed.” change to “… the differences in the expression of the genes which potentially are involved in regulation of reproduction were analyzed and discussed.”

Reply: Revised as requested(line115-116).

Comment 11: Pg 3 l 117-119. “The testis and ovary tissues were excised from D. hystrix individuals as soon as possible and quick-frozen in liquid nitrogen immediately, and then stored at -80°C until RNA extraction.” Change to “The testis and ovary tissues were excised from D. hystrix individuals within … from sacrifice, immediately quick-frozen in liquid nitrogen, and then stored at -80°C until RNA extraction.”

Reply: Revised as requested(line135-138).

Comment 12: Pg 3 l 117-119 ”… end (PE) reads with a length of 125 bp were generated.” Change to “… end (PE) reads of 125 bp length were generated.”

Reply: Revised as requested(line160-161).

Comment 13: Pg 3 l 136-137 “By means of SOAPnuke (version 1.5.0) [24], the raw sequencing data were quality- controlled with the parameters “-l 10 -q 0.5 -n 0.05 -p 1 -i”.” change to “By means of SOAPnuke (version 1.5.0) [24], the raw sequencing data were pruned using the software’s quality control with the parameters “-l 10 -q 0.5 -n 0.05 -p 1 -i”.”

Reply: Revised as requested(line163-164).

Comment 14: Pg 6 l 228-229 “… or not to be presented (0 ≤ FPKMs < 0.1)…” change to “… or not expressed at all (0 ≤ FPKMs < 0.1)…”

Reply: Revised as requested(line268-269).

Comment 15: Pg 13 l 395-397 “Thus, understanding the expression and endocrine regulation of steroidogenic genes would greatly help us establish better methods for controlling reproduction in D. hystrixa quaculture.” How it would help us in a practical aquaculture breeding terms? I understand that further research is needed to reach that level but an example of this kind of further research should be given.

Reply: An example has been added in the revised manuscript(line440-444).

Comment 16: Pg 13 l 416-418 “In the present study, the transcript levels of dmrt1 were determined to be significantly higher in the male gonads than in the female gonads, ovarian dmrt1 expression could not be detected by transcriptome analysis (Table 4).” It is confusing this sentence when is compared to the results. For example: In figure 5 with RT-PCR you present expression of dmrt1 but not in the ovary. From what it can be seen in the results, there is no significantly higher expression difference in male and female. In male is expressed but in females is not according to the RT-PCR. In the transcriptome analysis you mention that there is no expression detected. However in the results you mentions as DGEs the “doublesex- and mab-3-related transcription factor 1 (dmrt1)” and present it in Table 4.

Reply: Thanks a lot for your careful review. But we would like to clarify that, in figure 5 the, the y axis is log2 (Fold Change). Thus, it can be seen that both RNA-seq (blue) and RT-qPCR (red) results showed that dmrt1 was differential expressed (up-regulated in testis). Also, the RT-qPCR result was consistent with that of RNA-seq. The sentence ‘ovarian dmrt1 expression could not be detected by transcriptome analysis’ means that the FPKMs of dmrt1 gene in ovaries were ‘0’, because the mRNA level of dmrt1 in ovary is too low to be detected by RNA-seq.

Comment 17: Pg 13 l 478-481 “Hence, the expression profiles of the sex-biased genes identified in this study should be carefully analyzed in the future, and further molecular biology and genetic studies are required to clarify the sexual manipulation roles of these genes, and then confirm the sex determining gene.” A correct statement but it can be supported from a hypothetical example coming out from the results of this study could be given and discussed.

Reply: This section has been revised by adding a hypothetical example(line530-541). Thanks for your good advice.

Comment 18: Pg 15 l 484-486 “In this study, many genes associated with oogenesis, oocytes maturation, spermatogenesis and sperm motility were recognized, such as vasa,...” change to... “In this study, the expression of many genes associated with oogenesis, oocytes maturation, spermatogenesis and sperm motility was presented, such as vasa,...”.

Reply: Revised as requested(line545-548).

Comment 19: Pg 16 l 517-518 “In order to guarantee the performance of artificial propagation, more…” change to “In order to guarantee the performance reproductive management of this fish species under aquaculture conditions, more…”.

Reply: Revised as requested(line581-582).

We have tried our best to improve the manuscript and made the changes in the manuscript as your valuable suggestions. I hope that the revised manuscript represents a better version. Once again, special thanks to you for your warm works.

Reviewer 2 Report

General comments

The manuscript by Chen and colleagues, although has a quite simplistic experimental design (three male and three female comparison) the results may have some interest for the scientific community. The results are adequately presented and interpreted in the text;

Specific comments

Lines 86-87: Genes do not exhibit dimorphism but differential expression. Please rephrase

Lines 127-128: A simple agarose electrophoresis is not adequate to detect RNA degradation.

Lines 131-132 Q please provide further details on the library preparation kit in order for the experiment to be reproducible

Lines 142-146: Superfluous description (with inchworm, chrysalis and butterfly). Please remove it.  In any case that’s the way Trinity runs.

Lines 146-147: How redundancy was removed. Please explain.

Line 187: How b-actin was used as reference gene? Which were the criteria?

Line 189: Did you check for the qPCR efficiencies. Delta delta Ct method assumes you have equal or near equal PCR efficiencies. Please clarify.

Line 309: Which previous publication. This not the discussion section.

Line 323 What do you mean by interesting candidates. In this section you just have to present the results, but not to interpret them.

Author Response

Please see the enclosure file"Response to Reviewer 2".

Reviewer 3 Report

Manuscript titled "Comparison of gonadal transcriptomes uncovers reproduction-related genes with sexually dimorphic expression patterns in Diodon hystrix" is well written. The experiment was well designed and carried out. It seems that the increased interest in the described species may allow the development of its aquaculture. However, to make this possible, it is necessary to know his bioengineering of reproduction. The description of the specific sex markers expression, may allow the biotechnology of reproduction to be refined.

I only suggest that you remove from the discussion all references to Figures and Tables that are described in the results (e.g. lines 370, 451, 458, 488, 494).

Author Response

General comments: Manuscript titled "Comparison of gonadal transcriptomes uncovers reproduction-related genes with sexually dimorphic expression patterns in Diodon hystrix" is well written. The experiment was well designed and carried out. It seems that the increased interest in the described species may allow the development of its aquaculture. However, to make this possible, it is necessary to know his bioengineering of reproduction. The description of the specific sex markers expression, may allow the biotechnology of reproduction to be refined.

 Comment 1: I only suggest that you remove from the discussion all references to Figures and Tables that are described in the results (e.g. lines 370, 451, 458, 488, 494).

Reply: Revised as requested(line408-410,500-501,504-505,543-545,552-554).